# Arsenic Trioxide, Itraconazole, All-Trans Retinoic Acid and Nicotinamide: A Proof of Concept for Combined Treatments with Hedgehog Inhibitors in Advanced Basal Cell Carcinoma

**DOI:** 10.3390/biomedicines8060156

**Published:** 2020-06-11

**Authors:** Terenzio Cosio, Monia Di Prete, Elena Campione

**Affiliations:** 1Department of Dermatology, University of Rome Tor Vergata, Via Montpellier 1, 00133 Rome, Italy; terenziocosio@gmail.com; 2Anatomic Pathology Unit, University of Rome Tor Vergata, Via Montpellier 1, 00133 Rome, Italy; diprete.monia@gmail.com; 3Dermatology Unit, Department of Systems Medicine, University of Rome Tor Vergata, Via Montpellier 1, 00133 Rome, Italy

**Keywords:** arsenic trioxide, basal cell carcinoma, Hedgehog signaling pathway, itraconazole, nicotinamide, retinoic acid, smoothened receptors

## Abstract

The treatment of advanced basal cell carcinoma has seen a progressive evolution in recent years following the introduction of Hedgehog pathway inhibitors. However, given the burden of mutations in the tumor microenvironment and lack of knowledge for the follow-up of advanced basal cell carcinoma, we are proposing a possible synergistic therapeutic application. Our aim is to underline the use of arsenic trioxide, itraconazole, all-trans-retinoic acid and nicotinamide as possible adjuvant therapies either in advanced not responding basal cell carcinoma or during follow-up based on Hedgehog pathway. We have analyzed the rational use of these drugs as a pivotal point to block neoplasm progression, modulate epigenetic modification and prevent recurrences.

## 1. Introduction

Basal Cell Carcinoma (BCC) is the most common form of human skin cancer. Advanced BCC, including metastatic (mBCC) and local aggressive forms, is exceedingly rare with an estimated incidence of 0.0028% to 0.55%. However, it has been historically associated with a bad prognosis compared to other variants [1]. This evaluation has been revised by Wadhera et al., according to whom the lower estimation would translate to 1 in 35,000 patients which seemed too high, considering the total number of cases reported in the literature [2]. The median age of patients with mBCC at the time of diagnosis of the primary lesion is about 45 years, and metastases appear at a median of about 9 years later [1,3]. McCusker et al. reported cases of mBCCs, in a review from 1981 to 2011. Among 100 cases, median survival after mBCC diagnosis was 54 months (27 months for regional metastases and 87 months for distant metastases, respectively) [4]. Locally destructive tumors, which are typically associated with a very late presentation, are more frequent than mBCCs and may pose a significant therapeutic dilemma. Hedgehog (Hh) signaling has been implicated in the regulation of differentiation, proliferation, tissue polarity, stem cell population and carcinogenesis, especially in advanced BCC. Hedgehog signaling molecules in mammals include three ligands (Sonic hedgehog—SHH, Indian hedgehog—IHH and Desert hedgehog—DHH), two receptors (PTCH1, PTCH2), a key signal transducer smoothened (SMO) and three transcription factors, Zinc finger protein GLI (Gli1, Gli2, Gli3) [5]. The SMO function is physiologically inhibited by PTCH1 and PTCH2 until Hh ligands are not present. These events result in the nuclear translocation of Gli1 and Gli2 and the transcription of Hh responsive genes, through a still unclear intracellular pathway [6]. This pathway is physiologically responsible of the normal development of the embryo, but in patients with Gorlin-Goltz syndrome PTCH mutations cause uncontrolled signaling through SMO, with high rates of cancer, especially BCCs [7]. The identification of the first natural Hh inhibitor, cyclopamine, permitted to study extensively the pathway and led to concentrate to SMO binding as an effective way to downregulate the signaling. [8] As a consequence of the teratogenicity of cyclopamine, the research of Hh inhibitors with a better safety profile led to the development of vismodegib (GDC-0449) and sonidegib (NVP-LDE225) [9,10]. They are both approved as a new treatment option for recurrent and advanced BCCs [11]. Selectively inhibiting SMO, by binding different sites of the transducer, they regulate negatively the activation of the downstream Hh target genes [12]. Inactivating PTCH1 mutations lead to constitutive Hh pathway activity through uncontrolled SMO signaling. Targeting this pathway, vismodegib causes an impressive tumor regression in patients harboring these genetic defects [13]. Nonetheless, resistance in the tumor microenvironment has still been documented. Secondary drug resistive mutations in SMO have already been reported occurring after the beginning of anti-SMO therapy [14]. Mutations of SMO are responsible for up to 50% of advanced BCC cases resistant both to vismodegib and sonidegib [13,15]. Alongside SMO G497W mutation, that is responsible for primary resistance to vismodegib [13], competition binding assays identified other mutations causing a significant alteration in the binding affinity between SMO protein and Hedgehog pathway inhibitors. The most frequent is the D473A, which causes a drop in affinity for both vismodegib and sonidegib, while E518A determines a significant decrease in the binding affinity for vismodegib and a slight increase for sonidegib [16,17]. Furthermore, computational docking of vismodegib onto SMO protein revealed that the mutations W281, V321, I408 and C469 are localized near the drug-binding pocket and negatively affect the affinity of the drug for its molecular target [18]. Finally, changes in GLI1 gene copy number or loss-of-function mutations of suppressor of fusion (SUFU) have also been implicated in causing resistance to the therapy with vismodegib [17]. Furthermore, the most common adverse effects of vismodegib were mild to moderate and included muscle spasms, dysgeusia, decreased weight, fatigue, alopecia, and diarrhea. However, clinical studies documented a high percentage of therapy discontinuation by patients. For these reasons, we hypothesized that drugs, inhibiting Hh pathways at different levels, as arsenic trioxide (ATO), itraconazole (ITRA), all-*trans*-retinoic acid (ATRA) and nicotinamide (NAM) could bypass these resistances or could be combined as an adjuvant therapy in order to hit multiple metabolic stages of the same pathway (Figure 1 and Figure 2).

## 2. Putative Alternative Hedgehog Pathway Inhibitors

The examined drugs have three fields of application, concerning the need not only to eliminate the neoplasm, but also to avoid its relapse, through target mechanism of action (e.g., by inducing apoptosis), and to restore the field of cancerization. Herein we illustrate the rationale for the possible use of these four molecules, that intervene at various levels along the Hh signaling pathway, in association with the canonical Hh inhibitors (vismodegib and sonidegib). Completed or undergoing clinical trials concerning the treatment of advanced BCCs with these four molecules are reported in Table 1.

### 2.1. Arsenic Trioxide

Arsenic derivatives have been used for treatment of cancers and inflammatory diseases in traditional Chinese Medicine. In Eastern literature, interest on arsenic in leukemia was drawn by 1882 [19]. Arsenic trioxide was approved for the treatment of relapsed and refractory acute promyelocytic leukaemia (APL) in Europe in March 2002. It has been demonstrated that this drug is effective against all stages of APL, including relapsed cases or as first-line treatment. It is also useful in the consolidation/maintenance phase of treatment [20]. Arsenic trioxide is the most active and effective single agent available for therapy of APL [21]. Arsenic derivatives induce differentiation and apoptosis, inhibition of proliferation and show antiangiogenic effects [22]. Arsenic trioxide, in particular, prevents ciliary trafficking and destabilize GLI2, inhibiting Hh pathway downstream from SMO. Arsenic trioxide may bypass acquired mutations in SMO present in resistant tumors [23,24]. Combined treatment ATO-ITRA has an additive Hh inhibitory effect, which has proved to be effective in mBCCs [25]. Ally et al. reported a decrease of GLI1 mRNA expression by 75% from baseline in patients with mBCC treated with the combined regimen of ATO-ITRA (*p* < 0.001) [25]. Their study included five patients, who relapsed after SMO inhibitor therapy, in which 5 consecutive days of intravenous ATO were followed by oral ITRA, from day 6 to 28, for successive cycles. Three of the five subjects completed three cycle of treatment, while the others discontinued due to disease progression or adverse effects, as transaminitis, leukopenia, and infections [25]. Two of the patients involved in this study presented functional mutations at site D473 of SMO, which is known to alter the drug-binding pocket causing Hh inhibitors resistance [14], while another one had a germline polymorphism at site R168H [25,26]. No mutations were detected in the downstream Hh pathway genes SUFU or GLI1. The needing to find pharmacological combinations in order to bypass the resistance of the Hh pathway led Bureta et al. to the synergistic use of vismodegib and ATO, in association with temozolomide, in glioblastomas resistant to first-line therapy [27]. By their results, marked inhibition and decrease in tumor growth were observed in mice receiving combination therapy, unlike those getting single drug or vehicle treatment [27]. Thus, combination of ATO/vismodegib and temozolomide might represent an attractive treatment association highly effective on tumors. Arsenic trioxide side effects are leukopenia, increased serum urea nitrogen and creatinine levels, transaminases and dyspnea. Recent reports highlight a combined ATO-ITRA increase QTc interval. Therefore, care must be taken in cardiac patients or in patients taking other drugs that increase the QTc interval [28]. Jeanne et al. described in an ex vivo model the C212/213S mutant of the PML, which is critical for ATO binding [29]. Moreover, other two mutations (A216V and L218P) have been reported in ATO-resistant PML cases [30,31,32] and a mutational hot-spot domain (C212-S220) has also been described [33].

### 2.2. Itraconazole

Itraconazole is a triazole agent used to treat fungal infections, as candidiasis, aspergillosis, histoplasmosis, and in the prophylaxis in immunosuppressed patients. It induces a reduction of ergosterol, in fungi, and cholesterol, in mammals, mainly inhibiting lanosterol 14-α-demethylase. Recently, ITRA has been proved to be effective also in neoplasms therapy [34,35]. In cancer cells, ITRA could suppress activated SMO and GLI, inhibiting target genes, as SOX9/mTOR, cyclin D1 (CCND1), Wnt/β-catenin, Bcl-2/cyt C, PI3K/AKT/mTOR, vascular endothelial growth factor receptor 2 (VEGFR2), multidrug resistance protein 1 (ABCC1), resulting in a block of the growth and proliferation of many cancers in vivo and in vitro, arrest of the cell cycle, inhibition of the angiogenesis, and induction of the apoptosis and autophagy [28,36]. Itraconazole can block SMO receptor directly, acting on the top of the Hh pathway. The first significant breakthrough in understanding the role of SHH signaling in cancer progression was the discovery that mutations in the PTCH1 gene were responsible for Gorlin-Goltz syndrome [37]. Most patients tolerate well ITRA; the drug appears to be devoid of effects on the pituitary-testicular-adrenal axis. The most common side effects are related to gastrointestinal tract; rarely, transient increases in liver enzymes have occurred; however, no cases of symptomatic liver dysfunction have been reported. Sporadic cases of hypokalemia have been described [38]. Acquired SMO mutations, including SMO D477G, confer resistance to these inhibitors. Kim et al. reported that ITRA and ATO—two agents clinically used to inhibit Hedgehog signaling through mechanisms different from those of canonical SMO antagonists—retain inhibitory activity in vitro in all reported resistance-conferring SMO mutants and GLI2 overexpression. Itraconazole and ATO, alone or in combination, inhibit the growth of medulloblastoma and BCC in vivo, and prolong survival of mice with intracranial drug-resistant SMO D477G medulloblastoma [28,37]. A phase II, non-randomized clinical trial was conducted on 29 patients, 19 of whom were treated with ITRA. Two groups of patients, presenting more than one BCC larger than 4 mm in diameter, were enrolled: the first one received oral ITRA 200 mg twice daily for one month (cohort A); in the second one ITRA 100 mg were administered twice per day for an average period of 2.3 months (cohort B). Primary endpoint was a change in tumor proliferation and Hh activity, evaluated by Ki-67 index and GLI1 mRNA, respectively. Secondary endpoint consisted in tumor size changes. Itraconazole resulted to reduce cell proliferation by 45% (*p* = 0.04), Hh pathway activity by 65% (*p* = 0.03), and the tumor size by 24%. Four of the eight patients with multiple non-biopsied tumors achieved a partial response, while the other four had stable disease. Fatigue and congestive heart failure were the two adverse events recorded during ITRA treatment [39]. Nowadays, no data about tumoral resistance to ITRA are available, representing a first choice in a likely combination therapy.

### 2.3. Retinoids

Retinoids regulate gene transcription binding nuclear retinoic acid receptors (RAR) or retinoid X receptors (RXR). They can heterodimerize and recognize specific retinoic acid response elements (RAREs) in genes promoter regions. It has been reported that RA-treatment increases PTCH1 expression, through homeobox transcription factor Myeloid Ecotropic Viral Integration Site [40], with consequent inhibition of SMO and repression of the Hh signaling pathway. Its target genes include GLI1, PTCH1, CCND1, and cyclin E. RA-mediated induction of PTCH1 did not depend on SMO since their inhibitors do not prevent the induction [40]. Germline mutations in the PTCH1 gene are typical of all the BCCs in patients with nevoid BCC syndrome [41]. RA has substantial effects on epidermal growth and differentiation, and has been used for the treatment of many epithelial disorders. Goyette et al. have shown that RA can inhibit Gli activity in immortalized murine keratinocytes in an RA receptor-specific manner. This inhibition may occur—at least partially—through sequestration of the transcriptional coactivator cyclic AMP responsive element-binding protein and suggests a novel effect of retinoid excess on SHH signaling [42]. Combined ATRA-ATO oral therapy has been used for APL. It showed mild side effects and was well-tolerated by patients. The most common side effects are headache, dryness and gastrointestinal disorder [43]. A phase III blinded, randomized trial was conducted with a 2–6 years follow-up. A total of 1131 patients were enrolled and randomly assigned to two groups: 566 were treated with tretinoin, while the others received placebo cream. There were no side effects reported in the tretinoin group, even though 135 patients died during the study period. It was considered statistically significant, then the Central Human Rights Committee asked to stop the trial and continue only the follow-up of the patients until the end of the study [44]. Amino acid substitutions in the ligand-binding domain of RARα and PML-B2 domain of PML-RARα are described mechanisms of resistance to ATRA and ATO therapy, respectively [43].

### 2.4. Nicotinamide

Nicotinamide plays a key role in cell physiology, facilitating NAD-redox homeostasis and supplying NAD, as a substrate, to enzymes that catalyze non-redox reactions. Nicotinamide is a form of vitamin B3 and is fundamental in preserving genomic stability, having photoprotective and anti-inflammatory effects [45]. It is a versatile drug used in the clinical practice for the treatment of a wide range of dermatological diseases, including autoimmune blistering disorders, rosacea, atopic dermatitis, and acne [46,47,48]. High cellular turnover tissues, as skin, need high doses of NAD+ to balance genomic insults and intracellular redox stress [49]. If its level decreases, skin gets more sensible to sun-damage, as demonstrated in animal models, where NAD+ deficiency impairs DNA damage response and increases genomic instability with consequent skin sensitivity to UV radiation and cancer development [46,47]. Nicotinamide acts on p53 expression, inducing genomic instability and reducing cells survival following solar-simulated UV radiation [49]. Remarkably, older people, who are usually on multidrug therapies, some of which photosensitizer, frequently present NAD+ reduction, making them more prone to skin neoplasms. A tumoral more malignant phenotype is, furthermore, correlated with low NAD+ levels in the skin [47]. Thus, reducing genomic damage and optimizing DNA mechanism of repair, NAM may largely prevent skin carcinogenesis. As a substrate of poly-ADP-ribose polymerase (PARP), NAD+-dependent deacetylases of the Sirtuin family (SIRTs), and cyclic ADP-ribose synthases, NAD+ may regulate genome stability and cell senescence. Sharing a common substrate, those enzymes compete for NAD+ consumption [50,51,52]. SIRTs are NAD+-dependent deacetylases, which remove acetyl and succinyl groups from proteins. Numerous cellular processes are under SIRTs control, included transcription, mitochondrial biogenesis, inflammation, cellular genotoxic damages resistance, and metabolic dysfunctions [53]. SIRT1 acts a pivotal role in skin homeostasis, ageing, and carcinogenesis [53]. Increased NAD+/NADH ratio enhances SIRT1 action, resulting in higher deacetylation of its targets, including p53. On the contrary, low NAD+ level decreases SIRT activity. Transcription of several genes, involved in cell-cycle arrest, apoptosis, or DNA repair, are driven by p53; moderate SIRT1 activation inhibits p53, through its deacetylation, and prevents cellular apoptosis [53]. However, continuous SIRT1 activity induce cell death by accelerated NAD+ depletion [54]. A phase III, randomized, controlled trial, enrolling 386 Australians with at least two non-melanoma skin cancers (NMSC) in the previous 5 years, showed that oral treatment with NAM (500 mg twice per day for 12 months) safely and effectively reduced the development of new NMSCs and actinic keratoses (AKs). In particular, patients receiving NAM presented a reduction by 23% in both new BCCs and SCCs, compared to placebo (*p* = 0.02) [55]. We now have phase III level evidence of NAM’s ability to decrease the development of NMSCs and AKs in high-risk individuals [56]. The role of NAM compared to SIRT1 is still under discussion, but it has been postulated that NAM could increase cytoplasmic SIRT1 levels, inhibiting GLI1 and 2 [57]. Nicotinamide is generally well tolerated; minor side effects reported are stomach discomfort, nausea and headaches. It is well tolerated at high doses (1.5–3 g daily) for a range of clinical and research indications. No interactions have been found with ATRA, ATO and ITRA. Despite this, NAM also acts in photoprotection negating ATRA photo-sensibility [56].

## 3. Conclusions

In the next few years, we could witness an increasingly widespread use of polypharmacotherapy in the control of patients with advanced and mBCCs avoiding the onset of resistance, reducing the dosage of the single drugs and, consequently, their side effects. More trials are needed to verify efficacy, compatibility in the patient’s real-life and drug associations. In this proof of concept, we would propose the association of multiple molecules, combined with canonical Hh inhibitors in order to implement the therapeutic response, prevent the onset of resistance and reduce the dosage of individual drugs and related adverse effects.

## Figures and Tables

**Figure 1 biomedicines-08-00156-f001:**
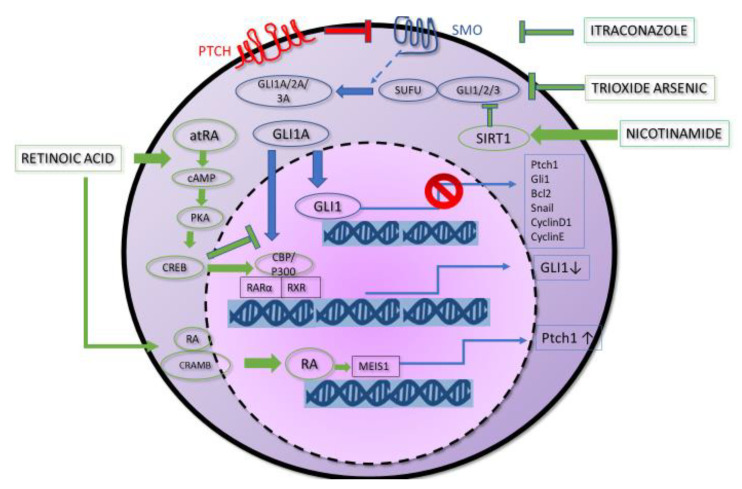
The different mechanisms of action of the different drugs are listed. Arsenic oxide inhibits Gli2 at cytoplasmic level. Itraconazole acts on SMO even in case of resistance to SMO inhibitors as it acts on a different site. Hence, the importance of using it in case of resistance, in a combined therapy with arsenic trioxide. All-*trans*-retinoic acid acts on two distinct ways: the first, dependent from RAR, allows the CREB bond by displacing Gli and thereby blocking its action on a nuclear level. The second, independent from RAR, occurs utilizing Myeloid Ecotropic Viral Integration Site (MEIS1), which leads to the transcription of PTCH1, leading to inhibition in the activation of the Hedgehog pathway. Nicotinamide works by increasing the levels of Sirtuin 1, which inhibits the Gli1 and Gli2 by acting in a parallel way to arsenic trioxide.

**Figure 2 biomedicines-08-00156-f002:**
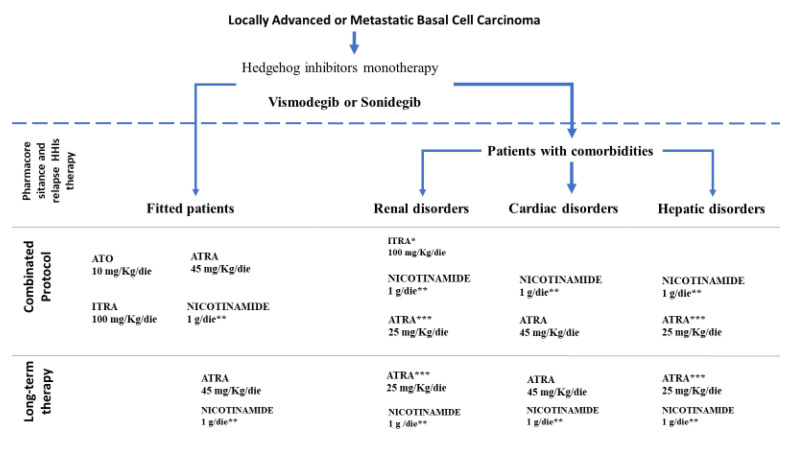
Example of treatment algorithms for advanced basal cell carcinoma. Due to disease heterogeneity in advanced basal cell carcinomas, actual treatment depends on tumor location, prior treatments, patient’s comorbidities and tumor mutation profile. In our rationale, we followed the position taken by a joint board between dermatologist, oncologist, surgeon and pathologist in order to give patients access to a polychemotherapy treatment. This kind of treatment acts immediately and favors the healing of areas with not clinically visible cellular atypia. These therapies administered orally could lead to maximum therapeutic adherence, thus maintain a good quality of life. At the same time, after the first treatment with sonidegib and vismodegib, the patient can continue the therapy adding itraconazole, arsenic trioxide, all-*trans*-retinoic acid and nicotinamide. The maintenance phase foresees the use of retinoids and nicotinamide, as also reported in the guidelines. * recommended dosage according to contraindications of itraconazole. Since it induces congestive heart failure, Itraconazole must be given in lower doses to patients with kidney disease. Itraconazole should not be used in the two weeks after interruption of treatment with CYP3A4 enzyme inducer drugs. ** dosage according to clinical trials, 1 g/die twice daily. *** recommended reduced dosage according to hepatic and renal disorders.

**Table 1 biomedicines-08-00156-t001:** The table summarizes the trials for each drug in the treatment of basal cell carcinomas.

Drug	NCT Number	Official Title on ClinicalTrials.gov or Publication Title	Phase	Sample Size	Study Results
itraconazole	NCT01108094 Completed February 2012	Pilot Biomarker Trial to Evaluate the Efficacy of Itraconazole in Patients with Basal Cell Carcinomas	II	29 pts Non-randomized Parallel assignment	Results available at https://clinicaltrials.gov/
	NCT02120677	A Pilot Study Investigating Antitumorigenic Potential of Topical Itraconazole in the Treatment of Basal Cell Carcinoma	Early I	5 pts Single assignment group	No results available.
Retinoic acid	NCT00005660 Completed November 2001	The Evaluation of Oral Acitretin in the Treatment of Psoriasis, Cutaneous Disorders of Keratinization, Multiple Basal Cell Carcinomas and Other Retinoid Responsive Diseases		130 pts	No results available.
	NCT00007631 Completed January 2009	CSP #402—VA Topical Tretinoin Chemoprevention Trial	III	1131 pts Randomized Parallel assignment	No results available.
Arsenic trioxide	NCT01791894 Completed June 2018	An Open-label, Biomarker Study of Arsenic Trioxide for the Treatment of Patients with Basal Cell Carcinoma	I II	5 pts Single group assignment	Results available at https://clinicaltrials.gov/
Nicotinamie	NCT03769285 Recruiting	Nicotinamide Chemoprevention for Keratinocyte Carcinoma in Solid Organ Transplant Recipients: A Pilot, Placebo-controlled, Randomized Trial	II	120 pts Randomized parallel assignment	No results available.

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
