# Peer review of "Arsenic Trioxide, Itraconazole, All-Trans Retinoic Acid and Nicotinamide: A Proof of Concept for Combined Treatments with Hedgehog Inhibitors in Advanced Basal Cell Carcinoma"

_biomedicines, 2020, doi:10.3390/biomedicines8060156_

Round 1

Reviewer 1 Report

The present review (in the first page is reported “article”) evaluated the use of arsenic trioxide (ATO), itraconazole (ITRA), all-trans-retinoic acid (ATRA) and nicotinamide (NAM) as possible adjuvant therapies in basal cell carcinoma (BCC), inhibiting Hedgehog (Hh) pathways at different levels. It is a well-written manuscript with well-organized figures and table, just some minor points need to be improved:

  • In the title, abstract and introduction, the Authors mentioned in order: arsenic trioxide, itraconazole, all-trans-retinoic acid and nicotinamide, to be more linear, the same sequence should be maintained in the detailed section 2.
  • In the Introduction section, the Authors should explain better the role and the action of sonidegib and vismodegib in the Hedgehog pathway.
  • In the section 2.2, it is important to add a brief description of the promyelocytic leukemia, in which the action of arsenic trioxide is o crucial.

Author Response

Dear reviewers, we have been impressed by your suggestions in improving the manuscript and by the ideas shared by you to carry out other works relating to tumour Biology and Pharmacological applications. In the resubmitted manuscript you can find all the corrections as required. We thank you for the time and the quality of your comments.

Reviewer 1

1)The present review (in the first page is reported “article”) evaluated the use of arsenic trioxide (ATO), itraconazole (ITRA), all-trans-retinoic acid (ATRA) and nicotinamide (NAM) as possible adjuvant therapies in basal cell carcinoma (BCC), inhibiting Hedgehog (Hh) pathways at different levels. It is a well-written manuscript with well-organized figures and table, just some minor points need to be improved: In the title, abstract and introduction, the Authors mentioned in order: arsenic trioxide, itraconazole, all-trans-retinoic acid and nicotinamide, to be more linear, the same sequence should be maintained in the detailed section 2.

1)Dear Reviewer, thank you for the notice. We have reversed sections 2.1 and 2.2, and the related refences, in order to make the text more orderly. 

2) In the Introduction section, the Authors should explain better the role and the action of sonidegib and vismodegib in the Hedgehog pathway.

2)introduction, line 43-54

The SMO function is inhibited by protein Patched (PTCH1, PTCH2) until Hh ligands are not present. These events result in the nuclear translocation of Gli1 and Gli2 and the transcription oh Hh responsive genes, through a still unclear intracellular pathway. [3] This pathway is physiologically responsible of the normal development of the embryo, but in patients with Gorlin-Goltz syndrome PTCH mutations cause uncontrolled signaling through SMO, with high rates of cancer, especially BCC. [Gorlin, 2004] The identification of the first natural Hh inhibitor, cyclopamine, permitted to study extensively the pathway and led to concentrate to SMO binding as an effective way to dowregulate the signalling [Cooper, 1998]. Cyclopamine resulted teratogenic, consequently the research of Hh inhibitors with a better safety profile led to the development of vismodegib (GDC-0449) and sonidegib (NVP-LDE225) [Robarge, 2009; Pan, 2010].

3) In the section 2.2, it is important to add a brief description of the promyelocytic leukemia, in which the action of arsenic trioxide is o crucial.

3) Dear reviewer, as suggested, we have introduced a specific focus on acute promyelocytic leukaemia in ex section 2.2 (now section 2.1). Line 116-125.

“Arsenic derivatives have been used for treatment of cancer and inflammatory disease in traditional Chinese Medicine. In Eastern literature, interest of arsenic in leukemia was drawn by 1882. (Doyle AC. Notes of a case of leukocythaemia. Lancet 1882; 119: 490.) Arsenic trioxide was approved for the treatment of relapsed and refractory APL in Europe in March 2002. It has been demonstrated that this drug is effective against all stages of acute promyelocytic leukemia, including for remission induction of relapsed cases, or as first-line treatment. It is also useful in the consolidation/maintenance phase of treatment. (Alimoghaddam K. A review of arsenic trioxide and acute promyelocytic leukemia. Int J Hematol Oncol Stem Cell Res.) 2014;8(3):44‐54.

Arsenic derivatives induced differentiation and apoptosis, inhibition of proliferation and antiangiogenic effects. Lengfelder E, Hofmann WK, Nowak D. Impact of arsenic trioxide in the treatment of acute promyelocytic leukemia. Leukemia. 2012;26(3):433‐442. doi:10.1038/leu.2011.245

ATO is the most active single agent available for therapy of APL and represents an efficient treatment option for all stages and age groups of APL(Zhou J, Zhang Y, Li J, Li X, Hou J, Zhao Y et al. Single-agent arsenic trioxide in the treatment of children with newly diagnosed acute promyelocytic leukemia. Blood 2010; 115: 1697–1702.) “

Reviewer 2 Report

The manuscript is well written and may be of interest to readers. Retinoic acid plus arsenic trioxide are seen as the ultimate therapy for leukemia [1-3]. Itraconazole and arsenic trioxide inhibit hedgehog pathway activation and tumor growth associated with acquired resistance to smoothened antagonists [4]. Effects of combined treatment with arsenic trioxide and itraconazole in patients with refractory metastatic basal cell carcinoma are described here [5]. Interestingly, increased concentrations of 3,4-didehydroretinol and retinoic acid-binding protein (CRABPII) in human squamous cell carcinoma and keratoacanthoma but not in basal cell carcinoma of the skin [6].

Authors may be interested on this catalog and they can find a myriad of different possible combinations for future possible therapies (and papers):

https://www.malacards.org/card/basal_cell_carcinoma?limit%5BMaladiesUnifiedCompounds%5D=329&showAll=False&limit[Publications]=1899

But one problem remains: testing

Some secondary observations:

This manuscript:

"Basal Cell Carcinoma (BCC) is the most common form of human skin cancer. Advanced BCC, including metastatic and local aggressive forms, is exceedingly rare with an estimated incidence of 28 0.0028% to 0.55%. However, it has been historically associated with a bad prognosis compared to other variants."

From Guidelines of care for the management of basal cell carcinoma:

"Basal cell carcinoma (BCC) is the most common form of human cancer, with a continually increasing annual incidence in the United States."

"Metastatic BCC is exceedingly rare, with an estimated incidence of 0.0028% to 0.55%, but has historically been associated with a very poor prognosis."

https://www.jaad.org/article/S0190-9622(17)32529-X/pdf

Moreover, this estimated incidence (0.0028% to 0.55%) is based on an article published in 1984. I presume that this incidence is not universal or timeless. Please see:

Metastatic basal cell carcinoma: report of five cases and review of 170 cases in the literature

The authors my wish to take a look at this article:

https://www.sciencedirect.com/science/article/pii/S0959804913011076#b0040

References

1. https://pubmed.ncbi.nlm.nih.gov/23841729/

2. https://pubmed.ncbi.nlm.nih.gov/23894153/

3. https://pubmed.ncbi.nlm.nih.gov/23017225/

4. https://pubmed.ncbi.nlm.nih.gov/23291299/

5. https://www.ncbi.nlm.nih.gov/pmc/articles/PMC4833646/

6. https://pubmed.ncbi.nlm.nih.gov/8618041/

Author Response

Dear reviewers, we have been impressed by your suggestions in improving the manuscript and by the ideas shared by you to carry out other works relating to tumour Biology and Pharmacological applications. In the resubmitted manuscript you can find all the corrections as required. We thank you for the time and the quality of your comments.

Reviewer 2

1)The manuscript is well written and may be of interest to readers. Retinoic acid plus arsenic trioxide are seen as the ultimate therapy for leukemia [1-3]. Itraconazole and arsenic trioxide inhibit hedgehog pathway activation and tumor growth associated with acquired resistance to smoothened antagonists [4]. Effects of combined treatment with arsenic trioxide and itraconazole in patients with refractory metastatic basal cell carcinoma are described here [5]. Interestingly, increased concentrations of 3,4-didehydroretinol and retinoic acid-binding protein (CRABPII) in human squamous cell carcinoma and keratoacanthoma but not in basal cell carcinoma of the skin [6].  

Authors may be interested on this catalog and they can find a myriad of different possible combinations for future possible therapies (and papers):

https://www.malacards.org/card/basal_cell_carcinoma?limit%5BMaladiesUnifiedCompounds%5D=329&showAll=False&limit[Publications]=1899

But one problem remains: testing

1) Dear reviewer, we are grateful to you for sharing your thoughts which will surely be a reason for investigation and possible proofs of concept to implement current therapies. The problem related to human test remains the greatest limitation, both in terms of construction and timing. However, being able to exploit some of the reported molecules (nicotinamide, ATRA) as adjuvant and unrestricted therapies, we hope to have the clinical counterpart in future.

About the other reflection, In vitro, tazarotene inhibited a murine BCC keratinocyte cell line, ASZ001, suggesting that its effect in vivo is by direct action on the actual tumor cells. Down-regulation of Gli1, a target gene of Hedgehog signaling and up-regulation of CRABPII, a target gene of retinoid signaling, were observed with tazarotene treatment. (So PL, Fujimoto MA, Epstein EH Jr. Pharmacologic retinoid signaling and physiologic retinoic acid receptor signaling inhibit basal cell carcinoma tumorigenesis. Mol Cancer Ther. 2008;7(5):1275‐1284. doi:10.1158/1535-7163.MCT-07-2043).

Some secondary observations:

1)This manuscript:

"Basal Cell Carcinoma (BCC) is the most common form of human skin cancer. Advanced BCC, including metastatic and local aggressive forms, is exceedingly rare with an estimated incidence of 28 0.0028% to 0.55%. However, it has been historically associated with a bad prognosis compared to other variants."

From Guidelines of care for the management of basal cell carcinoma:

"Basal cell carcinoma (BCC) is the most common form of human cancer, with a continually increasing annual incidence in the United States."

"Metastatic BCC is exceedingly rare, with an estimated incidence of 0.0028% to 0.55%, but has historically been associated with a very poor prognosis."

https://www.jaad.org/article/S0190-9622(17)32529-X/pdf

Moreover, this estimated incidence (0.0028% to 0.55%) is based on an article published in 1984. I presume that this incidence is not universal or timeless. Please see:

Metastatic basal cell carcinoma: report of five cases and review of 170 cases in the literature

The authors my wish to take a look at this article:

https://www.sciencedirect.com/science/article/pii/S0959804913011076#b0040

1)Dear reviewer, as suggest we read reported articles and correct epidemiological data.

We have read with interest the work of McCusker et al., and we have reported it in our manuscript. The article is also interesting in the new conception of locoregional cutaneous immunity and metastatization.

“This evaluation has been revised by Wadhera et al. according to with the lower estimate would translate to 1 in 35,000 patients which seemed too high, considering the total number of cases reported in the literature. [Wadhera A, Fazio M, Bricca G, Stanton O. Metastatic basal cell carcinoma: a case report and literature review. How accurate is our incidence data?. Dermatol Online J. 2006;12(5):7. Published 2006 Sep 8.] The median age of patients with metastatic BCC at the time of diagnosis of the primary lesion is about 45 years, and metastases appear at a median of about 9 years later.[1][ Ganti AK, Kessinger A. Systemic therapy for disseminated basal cell carcinoma: an uncommon manifestation of a common cancer. Cancer Treat Rev. 2011;37(6):440‐443. doi:10.1016/j.ctrv.2010.12.002 ] McCusker et al. reported in their review metastatic cases of BCC from 1981 to 2011. Among all 100 cases, median survival after mBCC diagnosis was 54 months, divided in 27months for regional metastasis ans 87 months for distal metastasis(McCusker M, Basset-Seguin N, Dummer R, et al. Metastatic basal cell carcinoma: prognosis dependent on anatomic site and spread of disease. Eur J Cancer. 2014;50(4):774‐783. doi:10.1016/j.ejca.2013.12.013).”

At last, we have noted that this is an Open review, but we do not fine Your signature. Following the interest and quality shown in its review, we make ourselves available for future collaborations or further suggestions.